# Fine-Scale Interactions between Leopard Cats and Their Potential Prey with Contrasting Diel Activities in a Livestock-Dominated Nature Reserve

**DOI:** 10.3390/ani13081296

**Published:** 2023-04-10

**Authors:** Chengpeng Ji, Hai-Dong Li, Wenhong Xiao, Kai Xu, Yingfeng Ren, Hongyun Li, Pengcheng Wang, Mingliang Fan, Xiaoqun Huang, Zhishu Xiao

**Affiliations:** 1State Key Laboratory of Integrated Management of Pest Insects and Rodents in Agriculture, Institute of Zoology, Chinese Academy of Sciences, Beijing 100101, China; 2University of Chinese Academy of Sciences, Beijing 100049, China; 3National Yugong Foresty of Jiyuan City, Jiyuan 454650, China

**Keywords:** site-use, temporal overlap, livestock disturbance, small rodents, camera trap

## Abstract

**Simple Summary:**

The spatiotemporal interactions between predators and their prey can be largely changed by livestock encroachment. Using camera trapping technology, we found that the two prey guilds with contrasting diel activities, nocturnal rats and diurnal squirrels, showed different habitat preferences with their predator, leopard cats (*Prionailurus bengalensis*). We also found that the fine-scale spatiotemporal use of leopard cats was consistent and highly correlated with that of nocturnal rats under livestock disturbance. Our results indicate that livestock disturbance could modify the site-use and temporal activities between leopard cats and their prey.

**Abstract:**

Habitat use and the temporal activities of wildlife can be largely modified by livestock encroachment. Therefore, identifying the potential impacts of livestock on the predator–prey interactions could provide essential information for wildlife conservation and management. From May to October 2017, we used camera trapping technology to investigate fine-scale spatiotemporal interactions in a predator–prey system with the leopard cat (*Prionailurus bengalensis*) as a common mesopredator, and its prey with contrasting activity patterns (i.e., nocturnal rats and diurnal squirrels) in a livestock-dominated nature reserve in Northern China. We found that the prey species showed different habitat preferences with the leopard cats. The nocturnal rats had strong positive effects on the site-use of the leopard cats, while the influence of livestock on the diurnal squirrels’ site-use changed from strong positive effects to weak effects as the livestock disturbance increased. The temporal overlap between the leopard cats and the nocturnal rats was almost four times that of the leopard cats and the diurnal squirrels, regardless of the livestock disturbance. Our study demonstrated that the fine-scale spatiotemporal use patterns of the leopard cats were consistent and highly correlated with the nocturnal rats under livestock disturbance. We suggest that appropriate restrictions on livestock disturbance should be implemented by reserve managers to reduce the threat to wildlife and achieve multi-species coexistence.

## 1. Introduction

Since predator–prey interactions are prevalent across different ecosystems, understanding the mechanisms driving their interactions are key issues in community ecology and conservation biology [1,2]. Ecological theories indicate that the predator–prey interactions often involve multi-dimensional dynamic processes. Previous evidence suggests that prey availability is more important than competition in determining the habitat selection and activity patterns of predators [3,4]. For instance, prey availability is likely the primary reason for the persistence of predators in plantations [5], and predators can also adjust their activity patterns to coincide with prey availability [6]. In many animal communities, the habitat utilization and diel activity of predators and their prey will also be determined by the environmental conditions (e.g., forest cover, roads, and settlements) [7]. In general, prey may actively partition habitat utilization to exploit areas shared with predators in a heterogeneous landscape [8]. Moreover, the trade-offs between avoiding a perceived predation risk and other fitness-enhancing activities in predator–prey systems, will be constrained and modified by the predatory cues, including those by human activity, since non-lethal human disturbance is comparable to those of natural drivers [9].

Growing evidence indicates that livestock encroachment can largely modify habitat use and the behaviors of wild animals [10,11]. However, different species may have divergent responses to livestock disturbance [12], since this disturbance may either strengthen or weaken the interactions between the predators and their prey. For example, livestock grazing can directly compete with native herbivorous mammals for limited food plants and space resources, forcing diurnal herbivores to shift their habitat use and temporal activities [13]. Moreover, they can indirectly act as super-predators and generate a non-lethal risk of predation particularly for larger carnivores [12], which ultimately weakens the trophic interactions between the predators and their prey. Generally, strong effects of predators on their prey can arise from the top-down effect if livestock encroachment could modify the habitat use of the top predators, while mesopredators may be attracted due to reduced predation risk, known as the mesopredator release hypothesis [14]. In addition, the predator–prey interactions may also be influenced by livestock grazing through the bottom-up effect. Small rodents are the main prey of mesopredators [4,15], but livestock activities, such as grazing, browsing, and trampling, can adversely affect the refuge and burrow availability for small rodents [16], consequently complicating the outcome of the predator–prey interactions. Furthermore, the spatiotemporal interactions between mesopredators and their prey can be largely changed due to human-induced disturbance and local population decline (even extinction) in large carnivores [17,18]. Thus, revealing how the interactions between mesopredators and their prey are influenced by livestock disturbance would be helpful to answer fundamental questions about how and why some species but not others, thrive in livestock-dominated landscapes.

The leopard cat, *Prionailurus bengalensis* from Family Felidae, is a typically nocturnal mesopredator in many temperate and (sub-)tropical forest ecosystems across most parts of Asia [19]. This cat can serve as an indicator species to prioritize conservation planning [20], due to the continued population decline as a result of habitat loss and illegal hunting [21,22]. This cat often uses opportunistic feeding strategies and exhibits plasticity in its habitat use and temporal activity to accommodate to habitat modification. For example, leopard cats may show a shift in habitat selection from their natural habitat to highly disturbed areas [23], and exhibit different diel activity among habitats and across seasons [24,25], including nocturnal-crepuscular activity [8,26], or even arrhythmic activity [27,28]. The plasticity of habitat use and temporal activity by leopard cats can optimize the exploitation of a diverse range of nocturnal and diurnal prey. Previous studies have shown that small nocturnal rodents are the main prey items of leopard cats, but their dietary composition may vary among different ecosystems due to the prey availability [15,26,29]. In Thailand, sciurid account for four percent and thirteen percent of the total diet of leopard cats in Phu Khieo WS and Huai Kha Khaeng WS, respectively, while murid feature 85% and 65%, respectively [27,30]. Moreover, the availability of small rodents may be affected by livestock disturbance if this disturbance could affect their habitat preferences and temporal activity patterns [31,32]. For example, diurnal squirrels associated with open habitats are found to show positive responses to grazing disturbance, while nocturnal rats associated with understory cover consistently show negative responses [33]. Thus, the above conditions could potentially strengthen or weaken the interactions between leopard cats and their prey with contrasting activity patterns. Although leopard cats have been found less spatially influenced by livestock disturbance [34], it is poorly understood whether and how livestock disturbance influences the interactions between leopard cats and their prey through the changed availability of prey. Therefore, illustrating the potential impacts of human disturbance on the interactions between leopard cats and their main prey can provide essential information for better understanding predator–prey interactions and the subsequent wildlife management [16,35].

We explore the potential effects of two prey guilds with contrasting diel activities (i.e., diurnal squirrels and nocturnal rats) on the spatiotemporal use of leopard cats as the main predators in a livestock-dominated nature reserve in Northern China (Figure 1). We tested the spatiotemporal effects of grazing intensity, and predicted that: (1) prey species might actively partition habitat utilization along spatial dimensions with their predators and livestock, since the two prey guilds select different habitats with the presence of either the leopard cats or livestock [8,36]; (2) the effect of the diurnal squirrels on the site-use of the leopard cats may be strengthened by the livestock disturbance, since diurnal squirrels prefer open habitats and would be positively affected by the livestock [33]; and (3) the temporal overlap between the leopard cats and the diurnal squirrels would increase, if diurnal squirrels reduced their daytime activity, but increased their crepuscular or even nocturnal activity in response to the livestock [13], which potentially would increase the encounter possibility with the leopard cats.

## 2. Materials and Methods

### 2.1. Study Site

This study was conducted in the Taihang Mountain Macaque National Nature Reserve (34°54′–35°42′ N, 112°02′–113°45′ E), Jiyuan city, Henan province, China. The reserve has a total area of 302.37 km^2^, with an elevation from 121 m to 1926 m. This area has a temperate continental monsoon; the dominant vegetation is warm temperate broadleaved deciduous forest, with an average annual precipitation of 646.4 mm, an average annual temperature of 14.3 °C, and snow cover usually lasting five or more months (from November to March). The original vegetation in this area was severely deforested before the reserve was established in 1998; the present vegetation types are dominated by *Quercus* species (Fagaceae), including *Q. variabilis*, *Q. aliena* var. *acutiserrata*, and *Q. baronii*. However, there are still some settlements and villages distributed in the reserve. Many roads were also built to facilitate the residents’ travel and forest fire prevention.

There are five native carnivores in the reserve: leopards (*Panthera pardus*), leopard cats, yellow belly skunks (*Mustela kathiah*), yellow-throated martens (*Martes flavigula*), and yellow weasels (*Mustela sibirica*). Domestic dogs fed by residents, as an invasive carnivore, were also easy to see in the reserve. Our study focused on the leopard cat as the key mesopredator, and we also considered the effects of other carnivores (native carnivores and domestic dogs) on the leopard cats. Although designated as a natural reserve, livestock such as sheep, cattle, and pigs are prominent within and outside the reserve. During the survey period, the sheep populations had at least 38 flocks with 20–250 individuals per flock, the cattle populations had at least 18 flocks with 5–17 individuals per flock, and the pig populations included at least 8 flocks with 5–17 individuals per flock. These livestock often grazed in the forests in the daytime and then were herded back into the intricate iron network distributed in the forests or barns at night. The grazing activities mainly occur from May to October.

### 2.2. Camera Trap Survey

In the reserve, we established 56 unbaited camera traps (LTL 6210 MC, Shenzhen, China) along roads and wildlife or livestock trails from May to October 2017 (Figure 2), where they were fastened to trees or bushes at a height of 0.3–0.6 m and programmed to take three photographs and video tips (10 s each) with a 0 s interval for each detected event. On average, adjacent camera sites were at least 300 m apart and the total number of cameras available allowed 56 camera stations to be operated simultaneously. The vegetation around them was cleared wherever necessary to avoid false triggering. All photographs, videos, and GPS information for each camera station were uploaded to the CameraData Network for Wildlife Diversity Monitoring (http://www.gscloud.cn/cameradata/)(accessed on 1 August 2018). Then, photographed or videoed animals were identified to the species level whenever possible. In this study, we obtained 7673 events of mammals and birds, but the other 563 events were not identified to species.

### 2.3. Covariates

In this study, we tested how the habitat and biotic covariates affected the species occupancy. The habitat covariates included elevation, slope, aspect, Enhanced Vegetation Index (EVI), forest cover, the distance to settlements, roads, and water stress (Appendix A) [10,19,37,38]. The quadratic effect of elevation and distance from settlements and roads was also tested. Relative abundance index (RAI) of other carnivores was calculated to reflect the probability of an encounter. A detailed description of covariates and data sources are shown in Appendix A. The percentage of forest cover around each camera site within buffer zones of 100 m, 500 m, and 1000 m were set to match the home ranges of nocturnal rats, diurnal squirrels, and leopard cats, respectively [25,39]. Since leopard cats were highly tolerated with the forest cover, we used 1000 m to reflect their fine-scale site-use.

Finally, the distances to both settlements and roads were used on the detection probability of our target species. The aspect was a categorical variable, and all continuous variables were standardized using *z*-scores. The degree of multicollinearity between variables was tested (Appendix A) using Pearson’s correlation coefficient [40].

### 2.4. Spatial Analysis

We used 14 days as a single sampling occasion for each camera site in the single-species occupancy models by using the unmarked package in R [41]. Occupancy was interpreted as the probability of site-use in our study, due to the larger home ranges of leopard cats and diurnal squirrels, which could violate the site closure assumption. We also calculated the naïve occupancy as the proportion of sites that recorded at least one photograph of the target species [42].

Three sequential stages were used for each species to identify the best covariates of detection and occupancy by employing a suite of habitat and biotic covariates: (1) detection model (Appendix A); (2) occupancy model (Appendix A); and (3) averaged single-species occupancy. These approaches enabled the modeling of species occupancy, and every covariate from the top models (ΔAIC ≤ 2.0) could be weighted to calculate the site-use for interspecific comparisons based on structural equation modeling (SEM). The covariates derived from the best occupancy models were identified, and all top-performing models were averaged to calculate the site-use for each species [43].

SEM was used to explore the direct and indirect relationships among the site-use of our target species [44]. However, we used livestock RAI at each camera site to reflect the intensity of livestock disturbance, since the most parsimonious models showed that the livestock site-use was not affected by any of the covariates. We categorized our camera sites as either higher or lower grazing sites by delineating the threshold using the mean value of livestock RAI, and built two models in the final SEM. The relationship between diurnal squirrels and nocturnal rats was not included in the final model. The maximum likelihood method was used for parameter estimation, and the overall fit of the SEM was evaluated based on Pearson’s χ^2^ test [45]. The strengths of interactions were classified as weak (≤0.14), moderate (0.15–0.50), or strong (≥0.51). All SEM analyses were conducted using the lavaan package in R [44].

### 2.5. Temporal Activity Patterns

The diel activity data were characterized by pooling the total number of independent detections across higher and lower grazing sites. Then, we fitted a kernel density function with the overlap package in R [46]. The sunrise (05:11) and sunset (19:29) times were used from the median date of this survey period (July 31, 2017). We defined the crepuscular hours of dawn and dusk as ±1 h from sunrise and sunset.

We performed pairwise comparisons of the activity patterns of our target species by using the overlap coefficient (∆) [46]. The coefficient ranged from 0 (no overlap) to 1 (complete overlap). Overlap indices were calculated using Dhat 4 (independent detections > 75) and Dhat 1 (independent detections < 50) equations according to the sample sizes obtained. Overlap coefficients and their respective 95% confidence intervals were calculated with the overlap package in R.

## 3. Results

We identified 5600 independent photos of mammal species from 8359 trap days across 56 camera-trap sites, including 259 detections of leopard cats, 410 detections of nocturnal rats, and 855 detections of diurnal squirrels. Livestock disturbances were also detected with 1817 events (Table 1). The leopard cats were widely distributed in the study area, with the highest naïve occupancy, followed by the diurnal squirrels, and the nocturnal rats. The RAI of the livestock was the highest, followed by the diurnal squirrels, but that of the leopard cats was the lowest (Table 1). The leopard cats and their prey were unevenly distributed, and mainly recorded in the northwest and east of our study areas (Figure 3).

### 3.1. Site Use and Driving Factors for the Leopard Cat and Its Prey

Based on the *β* estimates from the most supported single-species occupancy model, the most important factors all varied by species (Appendix A). The predicted site-use of the leopard cats increased with the distance to settlements (Figure 4a), whilst that of the nocturnal rats increased from the shady slope to sunny slope (Figure 4b) and that of the diurnal squirrels increased with the forest cover_100m (Figure 4c). The predicted site-use of the livestock was not affected by any variable (Figure 4d; Appendix A).

### 3.2. Direct and Indirect Effects on the Interactions between the Leopard Cat and Its Prey

The SEM met the criteria of Pearson’s χ^2^ test in the higher and lower grazing sites, respectively (χ^2^ = 0.291, *p* = 0.590; χ^2^ = 0.177, *p* = 0.674), indicating the model was closely fitted to the observed data. No significant missing paths were identified (all *p* > 0.05).

Overall, the interaction among our target species changed from the lower to the higher grazing sites (Figure 5). Specifically, the effect of the livestock on the site-use of the diurnal squirrels changed from significant and positive (Figure 5a) to non-significant and weak negative (Figure 5b) as with the increase of livestock disturbance. The effect of the nocturnal rats on the site-use of the leopard cats was significant and strengthened from the lower to the higher grazing sites. Several indirect effects on the leopard cat site-use were also identified (Appendix A).

### 3.3. Temporal Activity Patterns

The kernel density estimations of the daily activity rhythms revealed that the pattern for both the leopard cats and the nocturnal rats was “M” shaped, with two active peaks; however, both the diurnal squirrels and the livestock had one active peak. The leopard cats increased their dawn activity from 10.53% to 12.15% (proportion of detections of the target species at dawn) with the increase of livestock disturbance, and the diurnal squirrels increased from 1.94% to 3.80% (Appendix A).

The effect of the livestock on the activity of the diurnal squirrels and nocturnal rats was more evident, as the activity frequency of those species was much higher in the lower grazing sites (Appendix A). The temporal overlap between the leopard cats and their prey were both reduced from the lower to higher grazing sites (Figure 6a-b), and the temporal overlap between the livestock and the nocturnal rats was almost four times that of the leopard cats and the diurnal squirrels (Figure 6).

## 4. Discussion

Our results showed that livestock disturbance could change the effects of the two prey guilds of small rodents on the spatiotemporal use of leopard cats in the Taihang Mountain Macaque National Nature Reserve. We found that the effect of the nocturnal rats on the site-use of the leopard cats might be strengthened in the higher grazing sites, while the effect of the livestock on the diurnal squirrels was weakened. We also found reduced temporal overlap mainly between the leopard cats and the diurnal squirrels. These findings provide crucial information for our understanding of wildlife behavior and species interactions in livestock-dominated landscapes.

The distance to settlement, aspect, and forest cover, all had varying impacts on our target species, indicating that the prey species could actively partition site-use to co-occurrence with the leopard cats. We found that the naïve occupancy and estimated site-use of the leopard cats were very high, suggesting they had broader adaption across our study areas, especially at the sites further away from the settlement. The result was analogous to the studies that showed that the occupancy of leopard cats increased with the distance to settlements under livestock disturbance when the distance was less than 7 km [38]. Conversely, other studies, such as Mohamed et al. [47] and Wu et al. [19], found that leopard cats had a strong tendency to use and adapt to habitats near areas of human settlement. Whereas, wildlife will also shift their habitat selection due to habitat modification [23]. In the present study, we found the site-use of the diurnal squirrels and the nocturnal rats was strongly influenced by the forest cover and aspect, respectively. The studies by Feng et al. [38] and Reher et al. [39] also demonstrated that the food availability and microhabitat heterogeneity are the main factors to drive the habitat selection of small rodents. Nevertheless, some studies indicate that small rodents often use habitats near human settlements, since these habitats can be served as spatial refugia, food subsidies and reduce predation risks [7]. Free-ranging domestic dogs around settlements might be the reason to account for this distinction in habitat selection, as these dogs can also prey upon and compete with wildlife for food [48]. For example, Weng et al. [49] reported that the presence of domestic dogs had significant negative effects on the occurrence of leopard cats and other mammals; under such threats from dogs, the occurrences of native mammals tends to increase with the distance from human settlements [50].

We found that the effect of the livestock on the site-use of the diurnal squirrels and nocturnal rats was negatively affected in the higher grazing sites, especially for the diurnal squirrels. Some researchers suggest that the effects of livestock disturbance are heavier for trophic levels directly depending on plants and with closer dietary relationships [51], due to intense competition for limited resources [52]. *Quercus* species are the most common tree species in our study area. The seed crop (acorns/m^2^) of oaks was measured in our study areas from 2018 to 2021, mainly for *Q. variabilis*, there were: 37 acorns/m^2^ (N = 77 trees), 54 acorns/m^2^ (N = 117 trees), 27 acorns/m^2^ (N = 80 trees), and 43 acorns/m^2^ (N = 115 trees), respectively. These acorns are rich in lipids and proteins, therefore, they serve as the main food resource for small rodents, and are also foraged by livestock, which could create competition between livestock and small rodents [33]. We found the low level of livestock disturbance could promote the site-use of diurnal squirrels and nocturnal rats. This effect is likely mediated by the availability of invertebrate food sources for small rodents. Some studies have shown that low intensity of livestock disturbance can increase microhabitat heterogeneity and accelerate the leaf litter decomposition rates, which may be of benefit for invertebrates [53]. In addition, Schieltz and Rubenstein [33] demonstrated that ground squirrels preferred relative open habitats to improve their vigilance behavior. However, livestock intrusion in the present study area might habituate certain species to livestock grazing, possibly explaining the weak relationship between the livestock and the leopard cats [34,38].

The strong effects of the nocturnal rats on the site-use of the leopard cats might support the idea that livestock could drive out larger carnivores, and intensify the predation behavior of the mesopredator upon their prey [54]. Smith et al. [55] found that the number of preys killed by predators increased in disturbed areas. Furthermore, predator–prey interactions may also be highly dependent upon the prey availability [3]. For example, the occurrence frequency of prey species in the diet of leopard cats is found to vary significantly with elevation, with increasing occurrence of pikas and carnivores, but decreasing occurrence of rodents in higher altitude [29]. Furthermore, according to the optimal foraging theory, a predator should select prey with maximum energetic benefits in terms of size and availability [56]. In the study site, the food obtained from the diurnal squirrels was more effective and energy-saving, owing to their wider distribution and higher energetic benefits (they weigh approximately six times more than the nocturnal rats based on our live-trap survey (Ji, unpublished data)). However, we found that the leopard cats did not adjust their activity patterns and distribution to synchronize with the diurnal squirrels, likely suggesting that predators might have a certain preference for specific prey. Overall, the leopard cats showed considerable spatiotemporal overlap with the nocturnal rats, since murids are their main prey items in the diet of leopard cats in different ecosystems [15,24]. In our study area, additional work regarding the diet and foraging behavior of leopard cats, as well as the population dynamics of leopard cats and their prey, are needed to improve our understanding of their relationships with nocturnal rats and diurnal squirrels under livestock disturbance.

The present study showed that the diurnal squirrels, with their diurnal activity, might be more affected than the nocturnal rats, as the temporal overlap between the livestock and the diurnal squirrels was the highest for both the grazing sites. This indicates that the degree to which specific animals alter their activity patterns in the face of human disturbance is correlated with their natural activity patterns [13], as well as how they overlap temporally with human activity [57]. Compared with the spatial dimension, temporal avoidance may be more adjustable for prey species in adapting their activity patterns to limit synchronous activities with their predators or human disturbance [58]. Furthermore, we found the leopard cats reduced 17% of their diurnal activity and increased 15% of their dawn activity from the lower to the higher grazing sites. This suggests that the temporal activities of leopard cats may be much more flexible, and they are likely to reduce their diurnal activity and become increasingly nocturnal in a more disturbed area [24,25]. However, livestock disturbance could also aggravate temporal differentiation between leopard cats and diurnal squirrels, as revealed by their temporal overlap that reduced by almost 30% in the higher grazing sites. Puls et al. [59] found that goats (*Capra hircus*) comprised the highest biomass in the diet of leopards despite the fact they had the lowest temporal overlap. Therefore, caution must be applied when interpreting the current results, as only the broad activity patterns of the target species were included and site-scale temporal partitioning mechanisms used by prey and their predators under livestock disturbance were not addressed.

## 5. Conclusions

This research provided reliable evidence for the direct and indirect effects of livestock disturbance on the site-use between leopard cats and their prey. We found the effect of the livestock on the site-use of the leopard cats and their prey was weak, while the site-use and temporal activity of the leopard cats were highly dependent on the nocturnal rats. Given the importance as a priority area of biodiversity conservation in Taihang mountain, policies related to the appropriate restrictions on livestock grazing should be implemented, for example, converting free-ranging livestock to stall feeding or banning the livestock in the nature reserve. Overall, the livestock limitation could potentially benefit sympatric wildlife and their ecological functions and services in protected areas.

## Figures and Tables

**Figure 1 animals-13-01296-f001:**
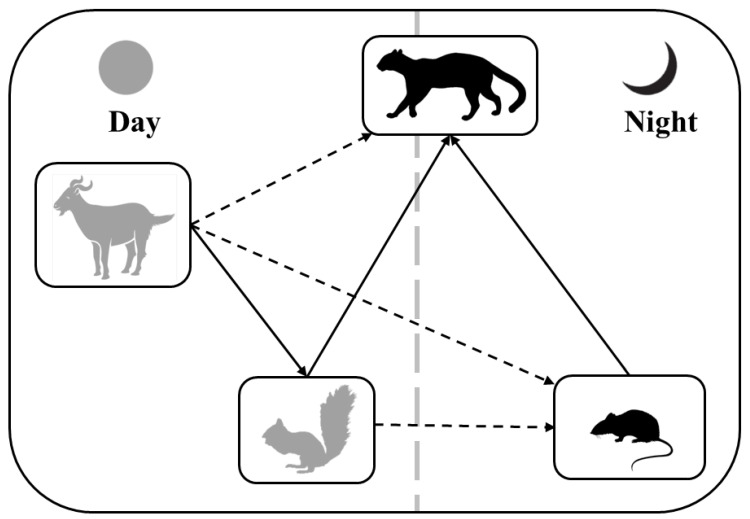
Conceptual model of the hypothesized interactions (e.g., site-use) between leopard cats and their prey with diurnal and nocturnal activity under livestock disturbance. Dashed arrows indicate negative effects, and solid arrows positive effects.

**Figure 2 animals-13-01296-f002:**
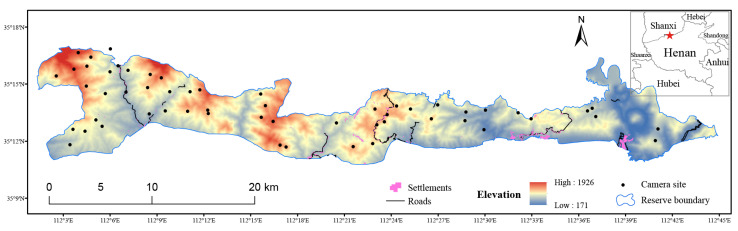
Study area and camera sites in the Taihang Mountain Macaque National Nature Reserve, Jiyuan city, Henan province, China.

**Figure 3 animals-13-01296-f003:**
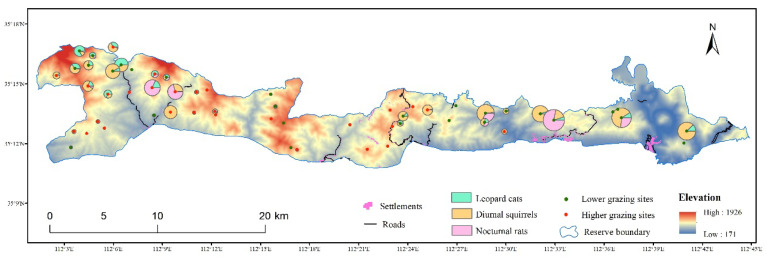
The composition of relative abundance index for leopard cats and their potential prey in higher and lower grazing sites in the Taihang Mountain Macaque National Nature Reserve, Jiyuan city, Henan province, China.

**Figure 4 animals-13-01296-f004:**
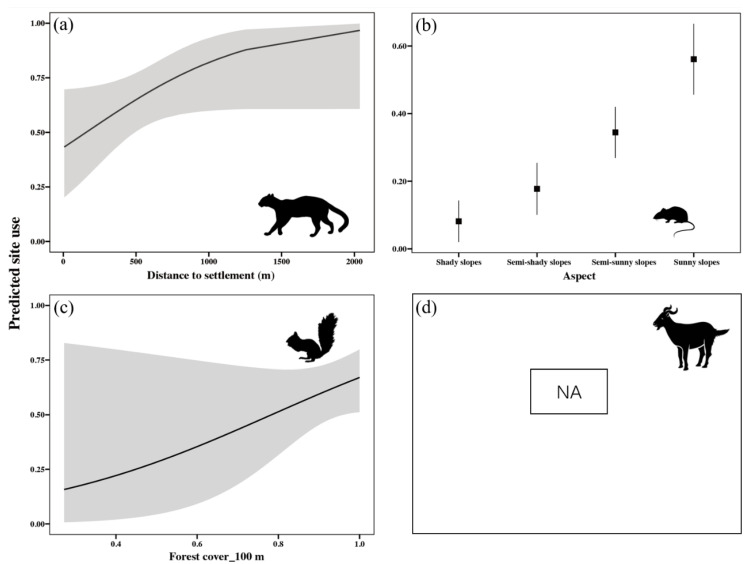
Predictions of the most important covariates influencing the site-use of leopard cats and their prey based on *β* estimates (solid line), along with their 95% confident intervals (grey shading), from each species’ most supported single-species model ((**a**) for leopard cat, (**b**) for nocturnal rats, and (**c**) for diurnal squirrels). The most supported occupancy model for livestock was the null model (**d**).

**Figure 5 animals-13-01296-f005:**
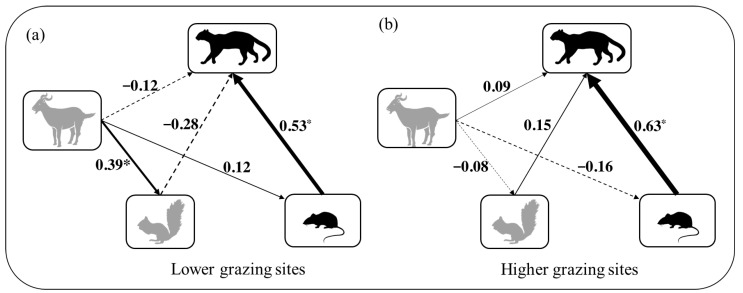
Net effects of standardized path coefficients of the SEM for leopard cats, their prey, and livestock in the higher (**a**) and lower grazing sites (**b**), respectively. Dashed arrows indicate negative effects; solid arrows represent positive effects. The numbers alongside the arrows are the standardized path coefficients. Paths with *p* < 0.05 are labeled with asterisk (*), and the line width is proportional to the size of the effect.

**Figure 6 animals-13-01296-f006:**
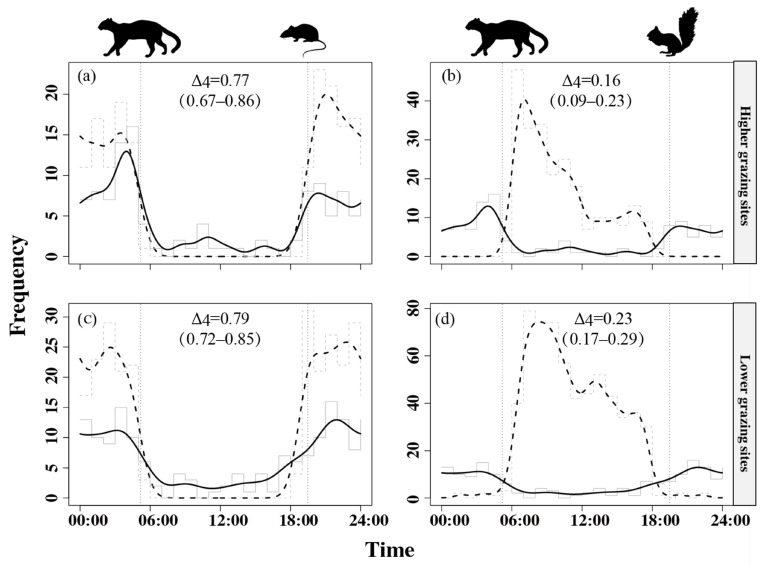
Pairwise comparisons of daily activity patterns between leopard cats (solid curves) and their prey (dashed curves) in higher (**a**,**b**) and lower grazing sites (**c**,**d**). The overlap coefficients (∆) and their respective 95% confident intervals are shown at the top of each graph. The vertical black dashed lines on the *x*-axis represent sunrise and sunset.

**Table 1 animals-13-01296-t001:** Numbers of independent photos and camera sites that recorded leopard cats, their potential prey, and livestock across 56 camera sites, along with their naïve occupancy and RAI.

Species	Activity Pattern	Number of Independent Photos	Camera Sites with Detections	Naïve Occupancy	RAI
Leopard cats	nocturnal	259	36	0.64	3.10
Nocturnal rats	nocturnal	410	18	0.32	4.90
Diurnal squirrels	diurnal	855	35	0.59	10.23
Livestock	diurnal	1817	38	0.46	21.74

## Data Availability

Research data are available when you contact the author.

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
