# Peer review of "Fine-Scale Interactions between Leopard Cats and Their Potential Prey with Contrasting Diel Activities in a Livestock-Dominated Nature Reserve"

_animals, 2023, doi:10.3390/ani13081296_

Round 1

Reviewer 1 Report (Previous Reviewer 2)

Thank you for amending this manuscript, which makes it much easier to read and interpret.

Author Response

Response: Thanks for your encouraging comments. We have also checked our English language. Please see the revised manuscript.

Reviewer 2 Report (Previous Reviewer 3)

While I'm not entirely satisfied, the revised manuscript has, in my opinion, been greatly improved. I still have some doubts about the English language, but I don't feel qualified to judge. After evaluation of the English language and editorial corrections, he recommends the manuscript for publication.

Author Response

Response: Thanks for your encouraging comments and patience. We have  included the description in the method part. Please see the revised manuscript.

Reviewer 3 Report (New Reviewer)

Comments on Manuscript "Fine-scale interactions between leopard cats and their potential prey with contrasting diel activities in a livestock-dominated nature reserve" Submitted to the Animals

General Comments

I appreciate the opportunity to review this exciting manuscript.

The manuscript investigates the spatial and temporal occupation of leopard cats, rats (murid), squirrels, and cattle in the Taihang Mountain Macaque National Nature Reserve. The author started from the premise that cattle are a proxy for human activity; and that the presence of cattle alters the interaction between prey and predator. The study lasted six months, recording images with camera traps. The results show a lot of interaction between rats and leopard cats at night but little interaction with squirrels. It seems that the presence of cattle modifies the interaction between mice and leopard cats more and less between leopard cats and squirrels.

The manuscript is original and deals with an important topic for managing and conserving wild species in that region. The findings are important for understanding how the presence of cattle in the natural reserve alters the interaction between prey and predator, with supposed general effects on the food chain in that region. The exposition of the study is well written, with clarity to understand the complexity of the findings. Below, I make some comments for parts of the text.

Simple Summary and Abstract

    The Simple Summary and Abstract are in clear, concise language and adequate for the depth level that the reader wants.

 Keywords

No suggestions.

Introduction

The introduction is well written, except for line 71, where the word "extinction" seems better than "expiration."

Material and methods

 Line 124: The description of the study area is poor. It remains to describe and show the settlements and main access routes on the map. These data increase the understanding of the study.

The Natural Reserve area is missing.

Line 126: Are no domestic dogs and cats in the nature reserve? Were they not caught on camera? As is known and the authors discuss in the text, dogs are important and growing meso predators in environments humans exploit (e.g., Hughes & Macdonald 2013). [Hughes, J., & Macdonald, D. W. (2013). A review of the interactions between free-roaming domestic dogs and wildlife. Biological Conservation, 157, 341-351].

Line 155: The covariates are elevation, slope, aspect, Enhanced Vegetation Index (EVI), forest cover, the distance to settlements, roads, and water stress. Since in temperate climate regions, the photoperiod is accentuated, and many species of animals have their activity strongly influenced by the duration of the photoperiod, I was surprised that this variable was not evaluated.

Results

The choice of statistical analysis was right, resulting in a comprehensive demonstration of the effects of interactions between species and covariates.

Discussion

The authors well discussed the complexity of the results. The interpretation is well-grounded in the scientific literature and theories that can support the discussion.

Conclusion

The conclusion is well articulated with the results and the discussion.

References

The cited references are within the scope of the study, but I found redundancy and excess. There is an excessive number of references (74), because several articles are redundant, as they deal with the same topic.

Correct the scientific names of animals in italics.

Author Response

Thanks for you time. Please see the attachment

This manuscript is a resubmission of an earlier submission. The following is a list of the peer review reports and author responses from that submission.

Round 1

Reviewer 1 Report

Dear authours,

I would delete "nature reserve" and "leopard cats" from Keywords, because they were already in the title. Furthermore, please check the references in the text. They do not always appears in journal citation style (numbers).

Furthermore, please consider to include absolute numbers (number of photographs) instead of relative index (density) in Figure 6 and Figure S2. It may be a better way to understand differences between high and low grazing sites.

Reviewer 2 Report

I did struggle to grasp the full meaning of this work. Whilst the grammar is excellent throughout, there are a number of areas where the sentence structure is not clear.

You use the term "rats" to refer to a number of different genera, including hamsters and voles (Line 155) and I'm not sure of the nocturnal activity of hamsters and voles (I certainly can catch Myodes and Apodemus during daylight). I think you need to clarify which species you are referring to, as your use of rats throughout confused me as to which species you meant.

In addition, I would like to see more detail on the camera trapping and results.  Clearing the vegetation may well reduce the frequency of small rodents in front of cameras - was the camera set to a high sensitivity to catch small mammals? Also the results appear low (less than one image per camera per day), particularly given the presence of livestock. Did you have any camera failures? Did any camera locations record no mammals at all?

In Figure 3c you give the relationship of aspect to squirrel activity, but aspect is circular, so 0 degrees should equal 360 degree and it does not. Also, please clarify if 0 degrees is north.

Figure 5 shows a number of relationship that are not statistically associated. High grazing appears to reduce squirrel activity (presence) as the only statistical change?

Figure 6 indicates very little change for rates but a reduction in temporal overlap with squirrels at high grazing sites. Does this imply it is less favourable habitat, or possible a small change in diet?

In the discussion you have the first mention of domestic dogs. Why was this not included in the analysis? Why are these 156 dog sightings not included in Table 2.

Detail of the above and other minor suggestions are in the pdf.

Reviewer 3 Report

It should be said that the authors undertook quite interesting research, which may have both application and practical significance. In the current environmental conditions, all behavioral analyzes of wild animals are of great importance for monitoring the species, threats and their scale, as well as can be valuable material for developing a protection strategy. In addition, the anthropogenic theme was raised in the evaluated work, which also affects the functioning of many populations of wild animals.

Some remarks came to my mind, which do not detract from the obtained results as well as the conducted research, but they should be clarified:

Line 106-110 is more of a methodology than an introduction?

In the introduction, the authors themselves write that the presence of livestock has a negative effect on the rat population. Therefore, it is worth emphasizing that in the methodology of the work, some values ​​(even approximate ones) should be provided regarding the abundance of these two species as a potential prey spectrum. And what about the spatial distribution, was it even, or and to what extent did it deviate from an even one? This is especially important in a situation where we are dealing with food opportunism, and this is the case with leopard cats, because it is the potential availability of prey that will mainly determine food behavior in the entire meaning of this phrase.

An equally important element is the assessment of the predation behavior of leopard cats in terms of their activity in obtaining food, whether it is still an innate (hereditary) activity or modified by anthropogenic elements.

I feel a clear lack of these two elements in the study.

Lines 143-145, in my opinion, a wrong approach, because despite the statement that the detection of other species of predators was less frequent, it does not mean that they did not play, perhaps, a significant impact on the elements of leopard cat predation. This should be clarified and developed, and possibly try to identify this impact.

Line 154, what justifies such a statement?

Line 154-159, what is the primary type of food after all - rats or squirrels?

Line 173-174, what was the recognition accuracy, what percentage of images were recognized?

The method part is way too long. It should be shortened and the most important information from the research should be provided, in particular the methodological assumptions and what they were conditioned by.

Line 289-300, I don't know if I understood correctly, it means one photo every 1.5 days (!)

In my opinion, there are far too many recorded leopard cats and too few of their prey, which may be the reason. Because with this data, the team of potential victims is very, very poor? Can this lead to any conclusions?

Line 300-301, what exactly does 1818 events with livestock recordings mean? Are these animals kept loose, in herds, what size are the groups, then it is necessary to explain why such a number of recordings?

In my opinion, the "results" presented in Figure S1 are not results in the sense of this study, but only a representation of the known behavior of the described species in terms of activity peaks. I am also not convinced that the grazing of livestock has a significant impact on this, these are species characteristics. This may be less true of squirrels.

Line 363-365, is not convinced of the 100% certainty of this conclusion, as other aspects related to the number and activity of rodents in this area have not been investigated, such as the reproductive potential in a given year, or the problem of dogs and cats in near human settlements. And the problem of homeless dogs (unless they don't exist there?) Coming back to reproduction, after all, from May to October - that is, during the research period, the breeding season lasts, and thus the number of victims, their activity and foraging activity increase?

Line 380 and 385, no need to list all the authors of this publication?

Line 415 and further, The authors write that oak was the most common species among vegetation and its fruits are food for small rodents, therefore, whether the assessment of seed generation during the study period was made, it is after all a basic factor for reproduction and survival of rodents, given the seasonality of Quercus fruiting.

Line 419-421, in my opinion unrelated to the study.

Line 421-423, is it really a competition for acorns, I don't know the specifics of livestock feeding in this area, but acorns do not convince me. I think this whole competition thread should be deleted.

Line 444, wrong quoting.

Line 446-450, in my opinion, this is not entirely true, because it depends on the behavior of cats, and not, as the authors suggest, energy costs. The biology of the species indicates that during the day when squirrels are active, cats are less active in obtaining food. In my opinion, this is a conclusion, not energy costs. Further, the authors write about natural patterns of activity and let it be mainly dependent. In general, I have doubts whether grazing will affect activity patterns as much as the authors suggest.

Line 464-470, first bad article citations. Secondly, the authors themselves state that there has been a shift in activity, and entourage activity is a species feature. So, was it so heavily dependent on grazing?

Line 480-483, in the light of earlier comments, is a bit exaggerated and one should not fully agree with it.

Although all the cases mentioned do not significantly diminish the very concept of the study, they may affect the results obtained, which is why, in my opinion, they require revision and refinement.

After a thorough explanation of the issues described by me and the correction of the text in the indicated fragments, I recommend the study for printing.

Round 2

Reviewer 2 Report

I remain confused as to your reporting of effects. Figure 4c refers to aspect, but since this is circular I don't see how this is a monotonic curve. Further, you make repeated reference to relationship that do not have any statistical significance. 

Reviewer 3 Report

Despite the corrections made by the authors, the manuscript is somewhat unsatisfactory.

Still, the authors did not provide the size or any approximate levels of density of the group of potential victims, and yet this is one of the most important issues in predator-prey interactions and is mainly dependent on the level of density of the predator and the group of victims. The mere statement "variability in the number and availability of local prey can lead to differences in the composition of leopard food" does not bring anything here?

The deletion of the wording - line 128-130 did not change anything.

Line 126-128 - is it still methodology?

Line 130-145 - these hypotheses are too long and you cannot fully agree with them, especially with 4, because in my opinion it is already a certain conclusion.

Line 163-169 - I'm not convinced

The methodology is still too long

Despite the explanation of 92.68% recognition of the number of camera traps mentioned by the authors, the number of findings seems too small, and thus the results are not very objective.

Line 373-374 - just deleting this information does not change the fact of my earlier assessment, it needs to be described somehow, not just get rid of the text?

Line 447-452 - same as above, deletion alone is not a solution.

I must also say that some of my previous comments have not been taken into account.

Although I like the concept of the study very much and it is worth publishing, the mentioned shortcomings and some of the previous review, in my opinion, require revision and refinement.

After a thorough explanation of the issues described by me, as well as those earlier, not fully (in my opinion, clarified) and the correction of the text in the indicated fragments, I recommend the study for printing.
